# Fast Emotion Recognition Based on Single Pulse PPG Signal with Convolutional Neural Network

**Min Seop Lee** [1], **Yun Kyu Lee** [1], **Dong Sung Pae** [1], **Myo Taeg Lim** [1,*], **Dong Won Kim** [2] **and Tae Koo Kang** [3,*]

1   School of Electrical Engineering, Korea University, Seoul 02841, Korea
2   Department of Digital Electronics, Inha Technical College, Incheon 22212, Korea
3   Department of Human Intelligence and Robot Engineering, Sangmyung University, Cheonan 31066, Korea
*   Correspondence: mlim@korea.ac.kr (M.T.L.); tkkang@smu.ac.kr (T.K.K.); Tel.: +82-02-3290-3243 (M.T.L.)

**Abstract:** Physiological signals contain considerable information regarding emotions. This paper investigated the ability of photoplethysmogram (PPG) signals to recognize emotion, adopting a two-dimensional emotion model based on valence and arousal to represent human feelings. The main purpose was to recognize short term emotion using a single PPG signal pulse. We used a one-dimensional convolutional neural network (1D CNN) to extract PPG signal features to classify the valence and arousal. We split the PPG signal into a single 1.1 s pulse and normalized it for input to the neural network based on the personal maximum and minimum values. We chose the dataset for emotion analysis using physiological (DEAP) signals for the experiment and tested the 1D CNN as a binary classification (high or low valence and arousal), achieving the short-term emotion recognition of 1.1 s with 75.3% and 76.2% valence and arousal accuracies, respectively, on the DEAP data.

**Keywords:** short term emotion recognition; one-dimensional convolutional neural network; PPG; personal normalization

## 1. Introduction

Emotions are conscious and/or unconscious feelings about a phenomenon or work, and are linked to mood, disposition, personality, and motivation. They can be expressed through various biological and physical reactions, including facial expressions, voice, text, gestures, and biosignals. Although human–computer interaction (HCI) has advanced, it remains challenging to identify human emotional states, which makes it difficult for machines to give the impression that they truly understand people [1–4]. Therefore, mutual sympathy between humans and machines is an important issue, and emotion recognition is a crucial element for HCI [5,6]. It is no exaggeration to say that considering human feelings can provide significant merits by inspiring confidence in modern complicated society. Moreover, detecting short moments of emotions is becoming important because emotion changes easily and quickly.

Emotional analysis has been investigated in various ways as technology has evolved, e.g., facial images, tone of voice, heart rate, etc. [7–9]. Among sensory data, physiological signals, such as electroencephalography (EEG), electromyograms (EMGs), photoplethysmograms (PPGs), respiration pattern (RSP), and electrocardiograms (ECGs) are key factors for emotion recognition, because these are spontaneous reactions, i.e., people cannot generally control these responses intentionally [10,11]. For example, if someone does not reveal their feelings through their facial expressions, i.e., "poker face", or does not say anything, it is difficult to detect their real emotion from their external appearance. However, their physiological signals, such as respiration or heart rate, are more specific,



because they are involuntary physical manifestations and contain considerable information about their emotional reaction.

Emotion recognition using physiological signals has been an active research field, developing many methods [12–14], but tending to two main approaches: classifying emotions based on hand-crafted features or deep learning frameworks to extract features. Kim et al. extracted many hand-crafted features from four-channel biosensors to classify emotions [15]. Giakoumis et al. proposed Legendre and Krawtchouk moments to extract features [16], and Wang et al. compared EEG features for emotion classification [17]. Jenke et al. reviewed feature extraction methods from EEG signals and selected features that were useful for emotion recognition [18]. Mert et al. proposed an emotion recognition method using multivariate extension empirical mode decomposition (MEMD) based on EEG signals [19]. However, methods using hand-crafted features had disadvantages, such as the recognition period being long and the accuracy being low.

As researchers began to use deep learning methods, many studies have employed deep learning frameworks as classifiers, e.g., Jirayucharoensak et al. proposed a deep learning model to identify feature correlation [20]. After deep learning method evolving, it has achieved remarkable outcomes in feature extraction and has been used to extract features from biosignals. Martinez et al. constructed physiological models to extract features using a convolutional neural network (CNN) and an auto-encoder [21]. Alhagry et al. proposed a long-short term memory (LSTM) model to learn EEG features and classify emotion depending on arousal and valence values [22]. Yang et al. introduced a parallel model of a recurrent neural network (RNN) and a CNN based on EEG signals to obtain meaningful features [23]. With deep learning, emotion recognition within one minute became possible.

This study employed plethysmograms (PPGs) to classify short-term emotions. PPGs measure changes in blood volume using an optical sensor, with the fingertip being the most reliable measurement position. We selected PPG signals because they contain rich information regarding human emotion [24], and since we want to recognize short-term emotion, we need efficient information extraction from less signal. PPG's shapes and values vary from person to person in different emotional situations and are commonly utilized for health problems. For example, if we watch a horrible scene or are in an embarrassing situation, our heart beats faster than normal, and we can measure blood pressure and stress levels using PPG [25]. Also, PPG signals are easily acquired using a small wearable device, with no need to attach patches to the body or large machines. For example, PPG sensor is embedded in existing devices such as smartphone (Samsung Galaxy) and smart watch (Apple Watch and Xiao Mi Band). A PPG signal is typically attached to the finger, therefore it can be obtained more easily than other signals. It is not comfortable to obtain and use the EEG signal which is the most common bio-signal in emotion recognition because devices for measuring EEG signals are expensive and the measurement process is cumbersome. Accordingly, many researchers have used PPG signals for the emotion recognition research [14,26–28]

Representing emotions as discrete classes is difficult, because emotions are mixtures of varying elements. Thus, we employed an arousal and valence emotion model to represent emotions using two parameters. We chose one-dimensional CNN (1D CNN) as the feature extractor for accurate emotional awareness. CNNs are widely used deep learning models for feature extraction and can handle 1D data well. We used a deep learning framework because we needed to extract many features with limited signal data.

We measured PPG signals for different arousal and valence values to establish the model was viable to classify emotions. The PPG signal was segmented into a single pulse as 1∼2 s to accomplish short term emotion recognition. We identified several trends for single pulse signal of different emotions and applied the model to various people. We used a database for emotion analysis using physiological signals (DEAP) emotion dataset to validate the model, which contains various physiological data about many people. We solved the problem of different signals from different people using a proposed personal normalization method. Experimental results show that single pulse of PPG signal can be useful for emotion recognition.

Section 2 describes the method to model emotions using two parameters and discusses limitations of previous emotion recognition methods based on hand-crafted features. Section 3 details the proposed 1D CNN method based on PPG signals. Section 4 introduces the DEAP dataset, experimental settings, and provide experiment outcomes. Section 5 summarizes and concludes the paper.

## 2. Related Works

### 2.1. Arousal Valence Emotion Model

It is not easy to model human emotions, since emotions are complex outcomes with many elements. Two methods have been commonly applied previously.

- Distinguish emotions as discrete labels, e.g., joy, sad, anger, happy, fear, etc. Although this method is conceptually simple, it is problematic when representing blended emotions that cannot be classified as a single case; and it cannot define the degree of emotion state, e.g., how glad you are.
- Use multiple dimensions to label emotions. However, this means each dimension is an emotional indicator, hence creating not a single scale but several continuous scales.

The most common emotional dimensions are valence and arousal, based on Russel's circumplex theory [29]. Valence (usually the horizontal axis) represents the degree of pleasantness, and arousal (usually the vertical axis) expresses activation level. Emotions were expressed in two-dimensional (2D) space, as shown in Figure 1, depending on their valence and arousal, e.g., anger has low valence and high arousal, whereas joy has high valence and high arousal. Thus, recognizing an emotion requires two binary classifications: high or low valence or arousal.

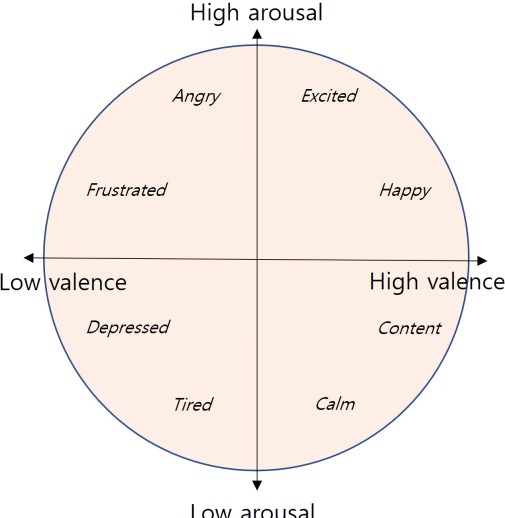

**Figure 1.** Arousal valence emotion model.

### 2.2. Hand-Crafted Features for Emotion Recognition

Initially, bio-signal features for emotion recognition were directly extracted using statistical calculations and frequency properties. More bio-signals were employed to extract more features for better performance. Kim et al. [15] obtained 110 features from four bio-signals (ECG, respiration, skin conductivity, and EMG), and selected hand-crafted features for different subjects and emotions among these features to represent emotions well. Hassan et al. used electro-dermal activity (EDA), PPG, and EMG signals to extract statistical features from the PSD of amplitude versus occurrence distribution [30].

As artificial intelligence evolved, deep learning frameworks began to be used in emotion recognition for classification. Previous manual features were used as inputs to the neural network.

Yoo et al. introduced an artificial neural network (ANN) to classify six emotions [26], using previous hand-crafted features from four signals as input to the ANN. Zheng et al. proposed the deep belief network (DBN) algorithm based on EEG signals [31], using manual features with DBN to classify emotions.

Although performance was improved, methods based on manual features have several limitations. They cannot represent signal details, i.e., inevitable information loss occurs when extracting features; and it is inappropriate to use hand-crafted features for short term emotion recognition because the features cannot be extracted quickly enough, statistical and frequency characteristics are meaningless for short periods.

Therefore, recent studies have investigated creating features using deep learning. Zhang et al. applied an auto-encoder to extract respiration signal features [32]. They tried to recognize short term emotions and perceived emotions for 20 s, achieving 73.06% accuracy for valence and 80.78% for arousal. Yang et al. extracted EEG features through the parallel deep learning model of CNN and RNN [23].

## 3. Short-Term Emotion Recognition with Single-Pulse PPG Signal

Figure 2 shows the overall structure for the proposed emotion recognition procedure. As discussed above, our purpose was to recognize short term emotions, which significantly reduces the amount of data available for feature extraction. Therefore, rather than attempting to use statistical features, we employed the raw PPG signal data. We preprocessed the raw data and segmented it as a single pulse, and then normalized the signals to reduce PPG signal size variations between people. This resulted in the single PPG signal pulse being expressed as a 140 × 1 vector, which was used as the input for the CNN. Section 3.1 discusses the reasons to use single pulse PPG signal by checking the subject's PPG pulse. Section 3.2 presents the proposed fast emotion recognition method using 1D CNN.

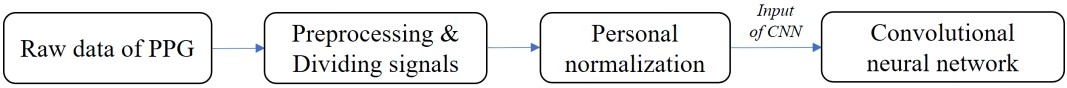

**Figure 2.** Emotion recognition procedure.

### 3.1. Single-Pulse Analysis of PPG Signal for Emotion Recognition

The PPG signal measures the periodic blood flow in blood vessel and incorporates considerable information regarding emotions. We employed the raw signal data to ensure fast emotion recognition, as discussed above and shown in Figure 3. We assumed that a single PPG pulse contained sufficient information regarding emotions, hence this raw data could be used to extract features.

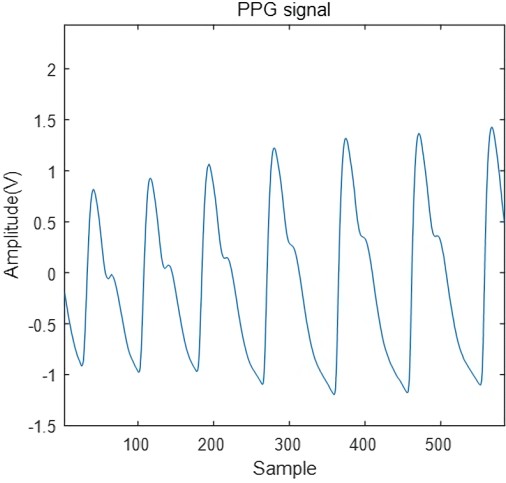

**Figure 3.** Raw photoplethysmogram (PPG) signal.

We observed PPG pulse shapes with different emotions. We selected four subjects between 20 and 30 to measure PPG signals, and chose each two music videos with high valence, low valence, high arousal and low arousal by surveying 10 people, following a similar method to that for the DEAP dataset. The music videos evoked emotions corresponding to valence and arousal, and we prepared a silent room to ensure the participants would not be disturbed by their environment. PPG signals were recorded from participant's fingers without them moving, to establish their neutral state, and then, while watching the selected music videos. Totally, we acquired the three classes data of PPG signals for valence and arousal (high, neutral, and low).

We split the PPG signals into a single pulse to check the tendency of the PPG signal with different valence, as shown in Figure 4 for one subject. The PPG single pulse data showed significant trends relative to valence and arousal for all subjects, e.g., pulses gathered together depending on the emotion. Similar pulses appear for similar emotions. Thus, we verified that emotions could be distinguished using a single PPG pulse, which could then be used as input for a deep learning framework such as CNN. The DEAP dataset included PPG data for a large group of people, which would provide good confidence in the proposed method outcomes.

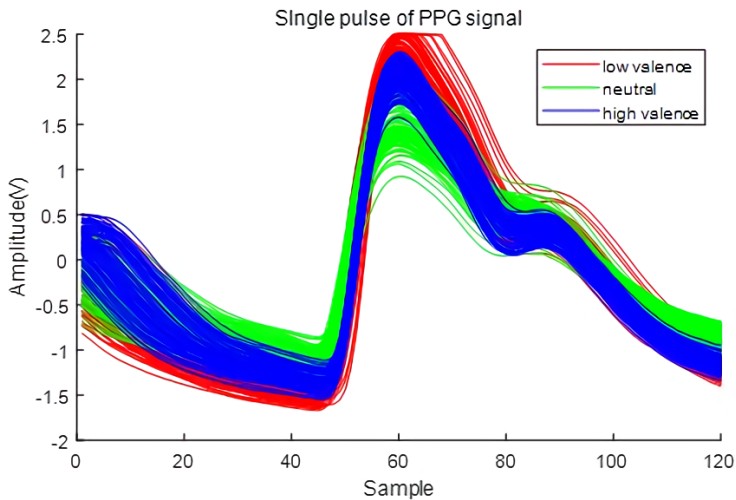

**Figure 4.** Dividing PPG signals into a single pulse.

*3.2. Feature Extractor Using Single-Pulse PPG Signal*

3.2.1. Dividing PPG Signals as Single Pulse

Measured signals can include patterns or trends that are not intrinsic to the data, which can hinder data analysis, and hence must be removed. PPGs are sensitive to disturbances such as the movement of the position where the sensor is attached. It is not easy for people to fix their fingers throughout the complete recording time, hence movements noise were generally included in the signal. PPG signals in the DEAP dataset included significant finger movement, as shown in Figure 5. Therefore, we preprocessed the PPG signals to eliminate movement trend before dividing the signals into single pulses.

We fitted a high-order polynomial to the PPG signal, and subtracted the fitted curve from the original PPG signal to remove the movement trend. We used an order of 50 polynomials to fit the original signal (for 60 s video clip). The preprocessed data was significantly more robust to finger movement than the original data, as shown in Figure 6.

After eliminating the trends due to the finger movement for PPG signal, we split the PPG signal into a single pulse. The PPG pulse always has a maximum value and we positioned the PPG signal maximum for each pulse at the center of the segmented signal. We used 1.1 s pulse length for optimal performance, corresponding to 140 samples for the DEAP dataset. The cutting window moved every

maximum peak and there were, on average, 20 overlapping samples between adjacent segments. There are approximately 120 samples between maximum peaks which means the signal's average period is 120 samples, and it is necessary to use overlapping samples for using all the raw data.

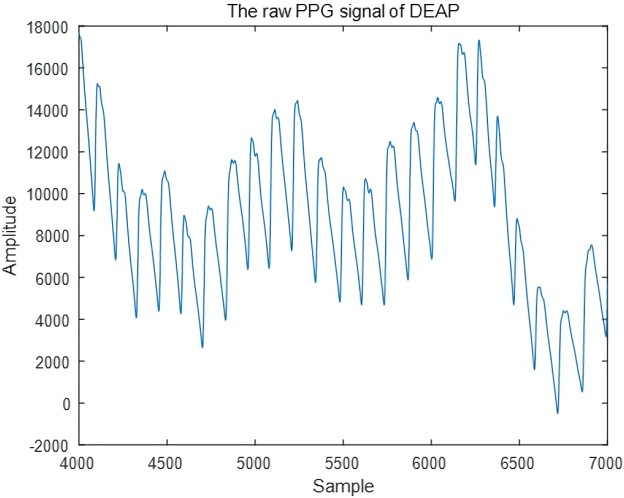

**Figure 5.** PPG signal of dataset for emotion analysis using physiological (DEAP).

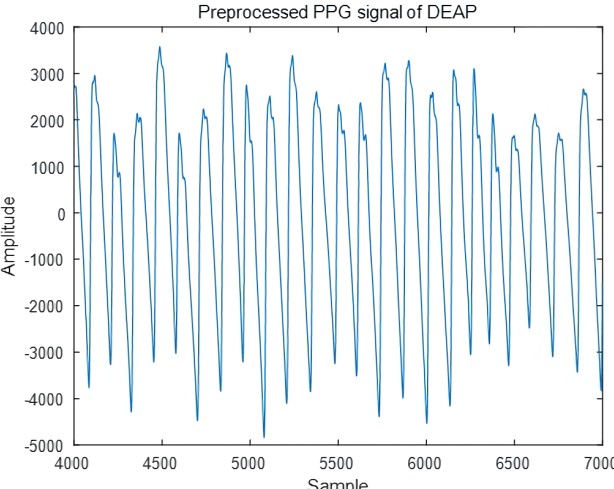

**Figure 6.** Eliminating finger movement noise for the DEAP dataset.

### 3.2.2. Personal Normalization

Normalization is essential when processing data that vary from person to person, such as biosignals. Generally, maximum and minimum PPG signal differed between the participants. Normalization method should be applied to compare the different people's signals, and of course we must be careful to preserve the signal's personal characteristics that differ depending on the emotions. Therefore, we normalized PPG signals using the personal maximum and minimum,

$$\bar{x}_i = \frac{x_i - min_{person}}{max_{person} - min_{person}} \times \alpha, \tag{1}$$

where $x$ is the original raw PPG signal, $\bar{x}$ is the normalized signal, and $\alpha = 1000$ to obtain signals with minimum = 0 and maximum = 1000, suitable for input to the 1D CNN. In this equation, we apply same minimum maximum value for same person. This simple normalization solved the problem with differences between people, with the normalized signal largely containing information regarding the signal shape rather than absolute value.

### 3.2.3. 1D-Convolutional Neural Network

Deep learning frameworks, particularly CNNs, are widely used for learning features from labeled data. CNNs have been shown to be effective feature extractors for various fields. Basic CNNs are composed of successive convolutional and pooling layers with a nonlinear function before pooling and fully connected classification layers. The final layer within the classification layers is commonly a softmax layer to score the class probability. The class is determined by a value with the greatest probability.

Figure 7 shows the proposed 1D CNN architecture to extract PPG signal features to detect the valence class. Network input was a preprocessed single pulse PPG signal which was a $140 \times 1$ vector. Since the physiological signal was a 1D vector, the convolutional operation was also a 1D operation. Feature extraction included two pairs of convolutional and pooling layers with activation functions, C1, C2, S1, and S2; where convolutional filters C1 and C2 were $3 \times 1$. The C1 layer contained 10 convolutional filters with stride 1 to allow padding, hence the C1 layer output 10 feature maps. Between the C1 and C2 layers, C1 outputs were subsampled in pooling layer S1 to reduce the features to only meaningful features. We used max pooling to return the maximum values in the $2 \times 1$ filters. The second convolutional layer, C2, contained 20 convolutional filters and the same process was repeated as in C1. To avoid overfitting, we introduced a batch normalization layer after each convolutional layer, which also resolved gradient vanishing and improved learning speed. Rectified linear unit (ReLU) was used as an activation function.

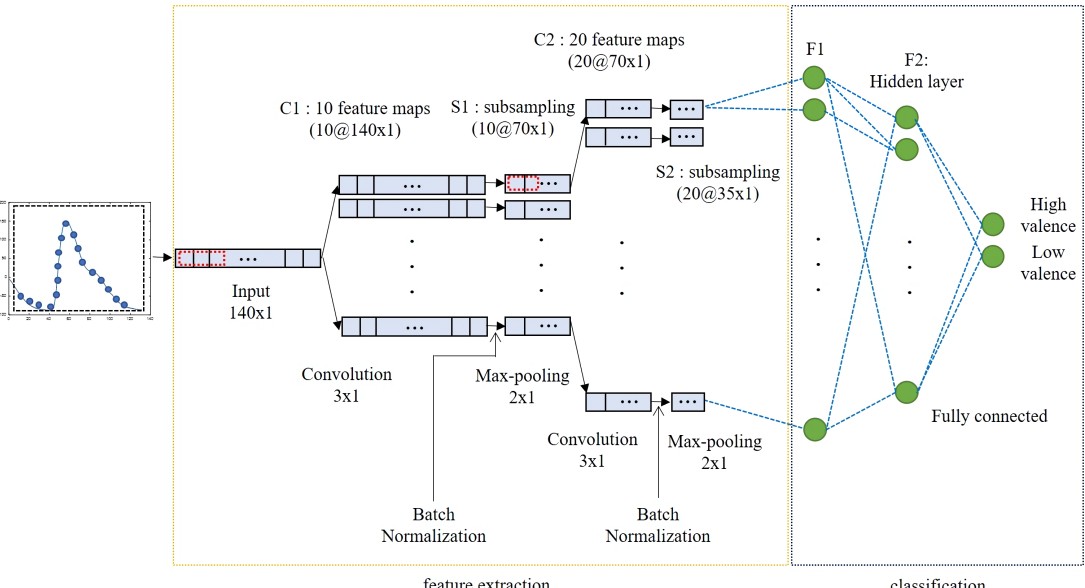

**Figure 7.** Proposed one-dimensional (1D) convolutional neural network (CNN) architecture for valence.

A total of 700 features were output from the second max pooling layer, F1. The hidden layer, F2, was formed by 600 nodes with a fully connected layer and ReLU layer was also used as a activation function. The final (classification) layer used the softmax to classify the level of valence (high or low). It calculates the probability of high valence and low valence. A higher probability value represents the level of valence (high or low). Likewise, the CNN model of the same structure with Figure 7 is applied to the arousal case that classifies the level of arousal (high arousal or low arousal).

## 4. Experiment

### 4.1. DEAP Dataset

To validate the proposed method, we employed the well-known DEAP dataset [33], downloaded from the DEAP homepage. DEAP contains various physiological signals, including PPG and others. There are 32 participants with different ages between 19 and 37 (50% female). Each participant watched 40 one-minute highlights of music videos that were preselected as visual stimuli to elicit emotions. All subjects were asked to rate each video by assigning values from 1 to 9 in terms of arousal, valence, dominance, like/dislike, and familiarity.

Since arousal and valence levels were 1–9, we divided the arousal and valence into two binary classes according to the threshold of 5 (high/low arousal and valence). The PPG data from the DEAP dataset was down-sampled to 128 Hz, and divided into a single pulse, as described above. Therefore, we obtained 55 segments from each one-minute music video clip, providing a total of approximately 22,000 data samples.

### 4.2. Experimental Setting

We trained the 1D CNN to extract features and classify emotions for the PPG single pulse signals extracted from the DEAP dataset. All the procedures were implemented in MATLAB R2018a. We applied the ReLU to the 10 feature maps from C1 and 20 from C2 after convolution and before pooling. Convolution stride = 1 and pooling stride = 2, indicating the degree of filter movement. The first and second fully-connected layers contained 700 and 600 neurons, respectively.

Dropout rate = 0.3, shuffled mini-batch size = 128, training epoch = 50, learning rate = 0.01, and we applied the cross-entropy cost function to train the CNN. Our model was trained using the optimization method of adaptive moment estimation (ADAM) for backpropagation algorithm. We used 80% of the DEAP dataset for training and the remaining 20% for test data.

### 4.3. Experimental Result

To test our neural network model, DEAP dataset was used for emotion recognition. Table 1 compares the proposed 1D CNN model with previous representative models using the DEAP dataset. We have 75.3% classification accuracy for valence and 76.2% accuracy for arousal. Also, we show the time values of recognition determine how fast the emotion recognition is possible because we have tried to perceive emotions at short intervals. The proposed model was able to provide short term emotion recognition, successfully detecting human emotions in 1.1 s.

This study proposed a short term emotion recognition method based on single PPG pulses. It is important to achieve high accuracy in a short period, hence we split the raw PPG signal into a single pulse that was input to a 1D CNN to extract features and classify emotions. The 1D CNN output was a binary classification indicating high or low arousal and valence.

In Table 1, various deep learning models, including DBN, CNN, auto-encoding, and CapsNet have replaced shallow methods, providing superior performance because deep learning can automatically extract features containing information that hand-crafted (statistical) features cannot access. In particular, the proposed 1D CNN model was superior to derive emotion related features from 1D biological signals.

The proposed 1D CNN method achieved short term emotion recognition, providing acceptably high accuracy compared with other methods for both arousal and valence, but only requiring 1.1 s which indicates the shortest interval of recognition. In particular, the proposed 1D CNN was more accurate and significantly faster than the latest EEG signal using CapsNet.

PPG signals include considerable information regarding emotions and can be easily measured using simple wearable devices, and the raw PPG signal is sufficient to represent emotions.

Consequently, fast emotion recognition could be implemented using the proposed 1D CNN architecture for single pulse PPG signals.

**Table 1.** Classification accuracy and recognition term for emotion recognition based on physiological signals.

| Method | Bio-Signal | Classification Accuracy | | Recognition Term |
|---|---|---|---|---|
| | | Valence | Arousal | |
| CNN (Martinez et al., 2013) [34] | BVP, SC | 63.3 | 69.1 | 30 s |
| SVM (Zhuang et al., 2014) [35] | EEG | 70.9 | 67.1 | 60 s |
| Hidden Markov models (Torres et al., 2014) [36] | RSP, GSR, EEG, TEMP | 58.8 | 75.0 | 60 s |
| Deep belief networks (Xu et al., 2016) [37] | EEG | 66.9 | 69.8 | 30 s |
| Multimodal deep learning (Liu et al., 2016) [38] | EOG, EEG | 85.2 | 80.5 | 63 s |
| Deep sparse auto-encoders (Zhang et al., 2017) [32] | RSP | 73.06 | 80.78 | 20 s |
| Multivariate empirical mode decomposition (Mert et al., 2018) [19] | EEG | 72.87 | 75.00 | 60 s |
| Multiband feature matrix and CapsNet (Chao et al., 2019) [39] | EEG | 66.73 | 68.28 | 3 s |
| Proposed 1D CNN | PPG | 75.3 | 76.2 | 1.1 s |

## 5. Conclusions

This paper investigated usage PPG signals for reliable emotion recognition. Arousal and valence theory, a typical method of modeling emotions, is adopted to classify emotions. For short term emotion recognition, the PPG signal was segmented as a single pulse which had a tendency depending on emotions and the personal normalization method is used to resolve the differences of PPG signal between the people. We applied a 1D CNN to extract features and classify emotions using normalized single pulse PPG signal. The final fully connected CNN layer incorporated the softmax function to assign high or low arousal and valence.

We used the DEAP dataset to verify that the proposed 1D CNN model achieved remarkable performance for classification accuracies and recognition interval. Thus, short term emotion recognition can be accurately achieved using normalized single pulse PPG signals. Our future work will attempt to use a combined model that shares the weights till convolution layers and applies different classification layers according to the emotional parameters. This work can generalize the model and save the training time.

**Author Contributions:** Conceptualization, M.T.L. and T.K.K.; Methodology, M.S.L. and D.S.P.; Software, M.S.L. and Y.K.L.; Validation, M.T.L. and D.W.K.; Formal analysis, M.S.L. and T.K.K.; Writing–original draft preparation, M.S.L.; Writing–review and editing, T.K.K. and M.T.L.

**Funding:** This research was supported by the National Research Foundation of Korea (NRF) funded by the Ministry of Education (NRF2016R1D1A1B01016071 and NRF-2016R1D1A1B03936281).

**Conflicts of Interest:** The authors declare no conflict of interest.

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
