# Peer review of "Fast Emotion Recognition Based on Single Pulse PPG Signal with Convolutional Neural Network"

_applsci, doi:10.3390/app9163355_

Round 1

Reviewer 1 Report

This paper is very well written with clearly presented results.

Author Response

We would like to thank for your valuable comments and suggestions. The paper has been revised by considering the comments and we denoted the modified parts.

Reviewer 2 Report

Dear Authors,

I find your work very interesting given the fact that such system could be used with less resources and hand-held devices such as health or fitness gadgets. I am not sure, but do you think the existing devices such as smart watches that can be used for heart rate measurement, those measure the level oxidation in blood using optical sensors, could provide you PPG signals?

The approach is good, however, I found some experimental flaws which drag my attention to neural network architecture. Fairly, proposed architecture is simple and claims of being fast recognition makes sense. However, the final output of the network, as you mentioned, uses softmax function (line no. 183 and 199). Softmax will return you the sum of all the probabilities thatequals to 1. That means, as you have two classes Valence and Arousal, it will always return you two values which sums 1. For example, you may have valence=0.1 and arousal=0.9 or other way around or valence=0.6 and arousal=0.4. You will always have one high and other low, and it is opposed to your Fig. 1. According to your reference figure Fig. 1 you will never get the outputs in other two spaces of the coordinates, (Low, Low) and (High, High), since you should have had three classes per dimension of valence and arousal, that you have acquired, line no. 144. In line no. 222 you said you use the cross-entropy cost function, which I suppose categorical cross-entropy as you have categories of the classes.

All those are doubts and seems like flaws to me, either representation/description or experimental. If it is representation error, you need a lot of improvements. If it is experimental flaw then you need to redo.

Thank you.

Author Response

Revision Reports

Manuscript ID: applsci-569005

Title : Fast Emotion Recognition Based on Single pulse PPG Signal with Convolutional Neural Network

We would like to thank the editor and reviewers for their valuable comments and suggestions. The paper has been revised by considering the comments and we denoted the modified parts with red letters. The detailed modifications in the revised version of the paper are as follows:

Answers to the reviewer #2’s comments

I find your work very interesting given the fact that such system could be used with less resources and hand-held devices such as health or fitness gadgets.

Q1. I am not sure, but do you think the existing devices such as smart watches that can be used for heart rate measurement, those measure the level oxidation in blood using optical sensors, could provide you PPG signals?

A1. Thank you for your comment. As far as we know, PPG sensor is embedded in existing devices such as smartphone (Samsung Galaxy) and smart watch (Apple Watch). These sensors detect the change in blood volume by illuminating the skin with the light from a LED and then measure the amount of light either transmitted or reflected to a photodiode. To mention specifically, we added examples of devices in the Introduction part as follow.

Before (Line no. 60)

Also, PPG signals are easily acquired using a small wearable device, with no need to attach patches to the body or large machines.

After

Also, PPG signals are easily acquired using a small wearable device, with no need to attach patches to the body or large machines. For example, PPG sensor is embedded in existing devices such as smartphone (Samsung Galaxy) and smart watch (Apple Watch and Xiao Mi Band).

Q2. The approach is good, however, I found some experimental flaws which drag my attention to neural network architecture. Fairly, proposed architecture is simple and claims of being fast recognition makes sense. However, the final output of the network, as you mentioned, uses softmax function (line no. 183 and 199). Softmax will return you the sum of all the probabilities thatequals to 1. That means, as you have two classes Valence and Arousal, it will always return you two values which sums 1. For example, you may have valence=0.1 and arousal=0.9 or other way around or valence=0.6 and arousal=0.4. You will always have one high and other low, and it is opposed to your Fig. 1. According to your reference figure Fig. 1 you will never get the outputs in other two spaces of the coordinates, (Low, Low) and (High, High), since you should have had three classes per dimension of valence and arousal, that you have acquired, line no. 144.

A2. Thank you for your comment about the model architecture. We used a softmax layer for the classification and it classified the two levels of valence as Fig. 7 (high valence/ low valence). Likewise, arousal classification is achieved through an independent model that has same architecture as Fig. 7 (high arousal, low arousal) which is mentioned in line no 200. Therefore, if model returns high valence=0.9 and low valence=0.1, it classified the emotion as ‘high valence’. Eventually, emotion classification is achieved through two similar models. In order to clarify for the arousal architecture, we modified the awkward sentence.

Also, the study conducted in Chapter 3.1 classified emotions as three categories using own PPG signal (high valence, low valence, neutral). This chapter was conducted to view trends in PPG signals according to the emotions and the dataset from this chapter was not the data used in the experiment in Chapter 4. We adopted the international emotion dataset for the experiment.

Before (Line no. 199)

The final (classification) layer used softmax to identify the valence and arousal. Since we wanted a binary classification, i.e., high or low valence and arousal, the same architecture was applied for arousal as shown in Fig. 7.

After

The final (classification) layer used the softmax to classify the level of valence (high or low). It calculates the probability of high valence and low valence. A higher probability value represents the level of valence (high or low). Likewise, the CNN model of the same structure with Figure 7 is applied to the arousal case that classifies the level of arousal (high arousal or low arousal).

Q3. In line no. 222 you said you use the cross-entropy cost function, which I suppose categorical cross-entropy as you have categories of the classes.

A3. Thank you for the comment about the cost function. It is suitable to use categorical cross-entropy for multiple classes. As we mentioned above, however, this study classified two categories of high value and low value. Therefore, we used the cross-entropy cost function

Reviewer 3 Report

Paper is good organized, conclusions are supported by results and compared with other results.

Author Response

(The authors gave the same response as above.)

Round 2

Reviewer 2 Report

Great now it seems clear what kind of architecture you are actually using. But in this case you need to have two copies of the same models for two parameters valence and arousal. I see this as a clear limitation of the architecture. As there are some work who shares some layers and use different output nodes.

Why wouldn't you simply use combined modelling for such classification? Where you share all your layers and use to output node for valence and arousal each with softmax. This way you may save some weight size and prevent double training of the same model.

Or wouldn't it be useful to share the weights till convolution layers and then apply different classification layers? This might also be useful to generalize the model for better performance.

The best way would be to use sigmoid function at the final two nodes for valence and arousal, they would give you a very high possibility to provide values from 0 to 1 for both the classes. It is just a suggestion but I found that a very powerful architecture to do this way. 

Thanks for the response. With best regards...

Author Response

Q1. Great now it seems clear what kind of architecture you are actually using. But in this case you need to have two copies of the same models for two parameters valence and arousal. I see this as a clear limitation of the architecture. As there are some work who shares some layers and use different output nodes.

Why wouldn't you simply use combined modelling for such classification? Where you share all your layers and use to output node for valence and arousal each with softmax. This way you may save some weight size and prevent double training of the same model.

Or wouldn't it be useful to share the weights till convolution layers and then apply different classification layers? This might also be useful to generalize the model for better performance.

A1. Thank for your valuable comments and suggestions. The method you mentioned is to use combined modeling that shares layers. This suggestion is efficient because it prevents double training and is useful for generalizing the model, however, it is not sure that classification is more accurate when using sharing layers rather than using independent neural network for valence and arousal. Nevertheless, your suggestion can be very useful. Therefore, we add your advice as a future work in the conclusion part as follow.

Before (Line no. 261)

Thus, short term emotion recognition can be accurately achieved using normalized single pulse PPG signals.

After

Thus, short term emotion recognition can be accurately achieved using normalized single pulse PPG signals. Our future work will attempt to use a combined model that shares the weights till convolution layers and applies different classification layers according to the emotional parameters. This work can generalize the model and save the training time.

Q2. The best way would be to use sigmoid function at the final two nodes for valence and arousal, they would give you a very high possibility to provide values from 0 to 1 for both the classes. It is just a suggestion but I found that a very powerful architecture to do this way.

A2. Thank for your valuable suggestion about the final layer. As you mentioned, sigmoid is widely used for the binary classification. We trained the model using sigmoid and softmax for the classification layer and we knew that the final classification layer didn’t significantly affect the training accuracy. In the future, we thought that valence and arousal could be classified by not two levels (high / low) but multi levels for representing detailed emotions, so we chose the last layer as a softmax.
